Review article

# Circadian and photoperiodic regulation of the vegetative to reproductive transition in plants
Fang Wang [1,3], Tongwen Han [1,3] & Z. Jeffrey Chen [2] ✉

As sessile organisms, plants must respond constantly to ever-changing environments to complete their life cycle; this includes the transition from vegetative growth to reproductive development. This process is mediated by photoperiodic response to sensing the length of night or day through circadian regulation of light-signaling molecules, such as phytochromes, to measure the length of night to initiate flowering. Flowering time is the most important trait to optimize crop performance in adaptive regions. In this review, we focus on interplays between circadian and light signaling pathways that allow plants to optimize timing for flowering and seed production in Arabidopsis, rice, soybean, and cotton. Many crops are polyploids and domesticated under natural selection and breeding. In response to adaptation and polyploidization, circadian and flowering pathway genes are epigenetically reprogrammed. Understanding the genetic and epigenetic bases for photoperiodic flowering will help improve crop yield and resilience in response to climate change.

Earth's orbit and rotation around the sun lead to cyclical changes in environmental factors, such as daily and seasonal changes in light, temperature, water, and nutrients, directly affecting plant growth and development, especially the transition from vegetative growth to reproductive development[1]. The sessile plants have evolved multiple strategies to anticipate these changes, synchronizing their life activities accordingly for survival and reproduction. Photoperiodism is a physiological reaction of plants to sense external environmental cues and to anticipate seasonal growth[2–4]. For many annual plants, the flowering time depends on the length of daily exposure to light. Based on whether their day length requirement is greater than the critical day length for reproductive transition, they are classified into long-day, short-day, and day-neutral plants[5].

The hourglass and the circadian rhythm models are two main hypotheses underlying mechanistic understanding of the photoperiodic response in plants[6–8]. The hourglass hypothesis assumes that the photoperiodic response of plants to short- and long-day conditions depends mainly on the length measurement of the dark period and the accumulation or activity change of specific chemical components[7]. This model can explain some photoperiodic events, but is contradictory to others when growing long-day plants under short-day conditions, a short amount of nighttime light can promote their flowering[9]. Thus, the hourglass model cannot fully explain all photoperiodic flowering phenomena. An alternative hypothesis

can better explain them; the circadian rhythm hypothesis emphasizes the interplay of circadian rhythms and light signals in the organism[10,11] and proposes key components in regulation of the light-controlled enzyme and substrate by the circadian clock. When the external light cycle synchronizes with the internal rhythm, the enzyme is activated by light and interacts with the substrate, inducing the expression of floral identity genes such as *CONSTANS* (*CO*) and *FLOWERING LOCUS T* (*FT*), leading to flowering transition[2–4,8].

The circadian clock regulates various transcriptional and post-transcriptional events at specific times of day, dating back to redox homeostatic mechanisms after the Great Oxidation Event at ~2.5 billion years ago[12]. This mechanism, known as circadian gating, adjusts an organism's sensitivity to environmental stimuli throughout the day. Circadian rhythm model of photoperiodic flowering is a good example. In Arabidopsis, precise daily control of *CO* expression by the circadian clock is essential for accurately measuring day length. During short winter days, *CO* peaks at night, leading to protein degradation. Conversely, in early summer's long days, *CO* peaks during daylight, stabilizing its protein and activating *FT* transcription for earlier flowering[13,14]. In addition to light, daily timing signals are provided by changes in temperature, which allow the clock to be entrained to run within a physiologically critical temperature range for a relatively constant amount of time. In plants, temperature

[1]State Key Laboratory of Wheat Improvement, College of Life Sciences, Shandong Agricultural University, Tai'an, Shandong 271018, China. [2]Department of Molecular Biosciences, The University of Texas at Austin, Austin, TX 78712, USA. [3]These authors contributed equally: Fang Wang, Tongwen Han. ✉e-mail: zjchen@austin.utexas.edu

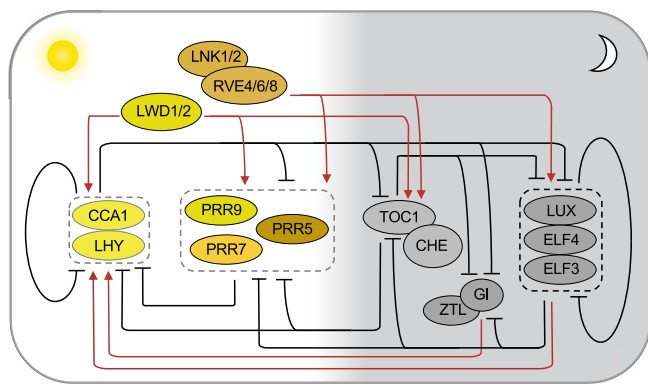

**Fig. 1 | Feedforward and feedback loops of the circadian clock in *Arabidopsis thaliana*.** The circadian clock in *A. thaliana* is regulated by a complex network of transcriptional feedback and feedforward loops that involve multiple clock genes. CCA1 and LHY are key transcriptional repressors that negatively regulate the expression of *TOC1*, *PRRs*, as well as their own transcription, and in turn, TOC1 and PRRs inhibit the expression of *CCA1* and *LHY*, creating a central feedback loop that helps regulate the clock and its output pathways. The evening complex (EC) formed by ELF3, ELF4 LUX, and GI, respectively, also regulates *CCA1* and *LHY* expression. RVEs are transcriptional activators that directly bind to the promoters of evening-phased genes repressed by CCA1 and LHY and promote their expression, thereby rest the internal oscillator. LNK1 and LNK2 serve as transcriptional coactivators via interaction with RVE to regulate target genes. LWD1 and LWD2 are both involved in perceiving light input to activate the central clock genes, including *CCA1*, *TOC1*, *PRR9*, and *PRR5*. White and gray shades indicate morning-phased (sun) and evening-phased (moon) events. Arrows and blunt ends show positive and negative regulation, respectively, while dashed boxes indicate proteins in a complex. For simplicity, protein and gene symbols are interchangeable in this and other figures.

responses and light signals are linked, and this emphasizes how complexly light, temperature, and clock genes interact to enable plants to synchronize their physiological activities with daily and seasonal cycles[15].

Notably, key flowering time controlling genes, such as *Flowering Wageningen* (*FWA*)[16,17] and *Flowering Locus C* (*FLC*)[18–21] in Arabidopsis and *CONSTANS-Like2* (*GhCOL2*) *Gossypium hirsutum* (Upland cotton)[22], are epigenes or epialleles, and their expression is regulated by DNA methylation, chromatin modification, and/or RNA-mediated mechanisms. Moreover, circadian clock gene expression is epigenetically regulated to promote growth vigor and defense in plant hybrids and allopolyploids[23–27]. In this review, we outline current progress in understanding the role of circadian regulation in the control of day-length-dependent flowering time in the model plant *Arabidopsis thaliana* and crops. Understanding circadian and epigenetic regulation of flowering time is crucial for improving crop yield and growth resilience, which may help develop gene-editing and epigenetic engineering technologies to optimize flowering times and yield potential and stability to meet the growing demand for food, feed, fuel, and biomaterials.

## Transcriptional architecture of the circadian clock in Arabidopsis

In Arabidopsis, the circadian clock network responsible for generating rhythmic outputs has been dominated by repressive feedback loops (Fig. 1). The central loop of core oscillators consist of two dawn-phased transcription factors *CIRCADIAN CLOCK-ASSOCIATED1* (*CCA1*) and *LATE ELONGATED HYPOCOTYL* (*LHY*), and their reciprocal regulator *TIMING OF CAB EXPRESSION1* (*TOC1*), aka, *PSEUDO-RESPONSE REGULATOR1* (*PRR1*), whose expression peaks in the evening; they constitute the central loop of the core oscillator[28–30]. Another factor CCA1 HIKING EXPEDITION (CHE), a TCP transcription factor, suppresses *CCA1* suppression through interaction with TOC1[31]. The *TOC1* homologs *PRR9*, *PRR7*, *PRR5*, and *PRR3* are sequentially expressed throughout the day and show partially redundant functions in repressing *CCA1* and *LHY*

transcription from dawn to dusk; they form an additional regulatory circuit with *CCA1* and *LHY*[1,32]. In turn, *CCA1* and *LHY* transcription represses expression of *PRR5*, *PRR3*, *TOC1*, *GIGANTEA* (*GI*), and the evening complex (EC) that comprises *EARLY FLOWERING3* (*ELF3*), *ELF4*, and *LUX ARRHYTHMO* (*LUX*)[33–35], but activates *PRR9* and *PRR7* expression in the morning[36]. In the evening, the EC functions as a transcriptional repressor to inhibit *PRR9* and *PRR7* expression, leading to the release of *CCA1* and *LHY* from repression[33,37]. Moreover, *GI*, a plant-specific protein gene that is repressed by CCA1 and LHY complex in the morning and by TOC1 and EC in the evening, induces *CCA1* and *LHY* expression through an unknown mechanism[30,38,39]. GI is localized in the cytosol where it physically interacts with a light photoreceptor ZEITLUPE (ZTL) with E3 ubiquitin ligase activity to mediate proteasomal degradation of PRR5 and TOC1[40,41].

In addition to the negative interactions, multiple transcriptional activators are found to complement the plant circadian regulation. *LIGHT-REGULATED WD1* (*LWD1*) and *LWD2* are part of a larger family of proteins known as WD-repeat proteins, which are involved in transmitting light signals to the circadian clock to promote expression of *CCA1*, *TOC1*, *PRR5*, and *PRR9* in the morning[42–44]. The midday-phased components REVEILLEs (RVE4, RVE6, and RVE8) are considered to be homologs of CCA1 and LHY due to their shared structural and functional similarities[45]. Interestingly, RVEs antagonize the action of CCA1 and LHY and bind to the same evening element (EE) during the midday period. This competition results in the activation of clock genes such as *PRR9*, *PRR5*, *TOC1*, *GI*, *ELF4*, and *LUX*. In turn, PRR5, PRR7, and PRR9 repress *RVE8* expression, indicating that RVEs play an important role in maintaining the balance and robustness of the circadian regulation[45,46]. Two other activating components NIGHT LIGHT INDUCIBLE AND CLOCK-REGULATED1 (LNK1) and LNK2 are transcriptional coactivators of RVE8 to regulate expression of *PRR5* and *TOC1*, both of which repress *LNK1* and *LNK2* expression in return, forming another loop[47].

## Circadian-dependent *CO* accumulation regulates photoperiodic flowering in Arabidopsis

Arabidopsis is a facultative long-day plant with metabolic pathways adapted to different photoperiods for flowering, and the control of its flowering timing involves the circadian clock, light signaling pathways, and transcription factor networks (Fig. 2). Among these, *CO* is one of the key genes that control the photoperiodic flowering response. In long-day conditions, *CO* transcripts accumulate from the afternoon into the night, which leads to the stabilization of CO protein and the activation of *FT*, a downstream gene that promotes flowering[48]. In short-day conditions, *CO* transcription is restricted to the dark, and *FT* expression is not activated. CYCLING DOF FACTORs (CDFs) are a family of transcription factors and can directly bind to the promoter region of the *CO* gene and repress its transcription in the morning[49,50], and this repression is necessary to ensure that flowering occurs at the appropriate time in response to seasonal changes[8,51]. Natural variations in CDF binding sites are correlated with differences in *CO* transcript abundance and flowering time[52]. Thus, precise control of daily *CDFs* expression and CO protein abundance is crucial for the proper regulation of flowering timing and adaption to changing environmental conditions.

In Arabidopsis, the transcription of *CDFs* is regulated by various clock components. Specifically, *CDFs* are induced by the morning clock genes *CCA1/LHY* and repressed by the evening clock genes *PRR5*, *PRR7*, and *PRR9*[32]. This leads to a typical diurnal expression pattern of *CDFs* with a peak at dawn and this regulation is essential for plants to optimize their reproductive success by flowering at the appropriate time of year[53–56]. Loss-of-function mutations in *CCA1* and *LHY* lead to early flowering[29], while constitutive expression of *CCA1* leads to circadian arrhythmicity and late flowering[28]. Additionally, the evening clock genes *PRR9*, *PRR7*, and *PRR5* function antagonistically with *CCA1/LHY* and coordinately and positively regulate CO dependent photoperiodic flowering process[53]. CCA1/LHY also represses the transcription of *CO* by

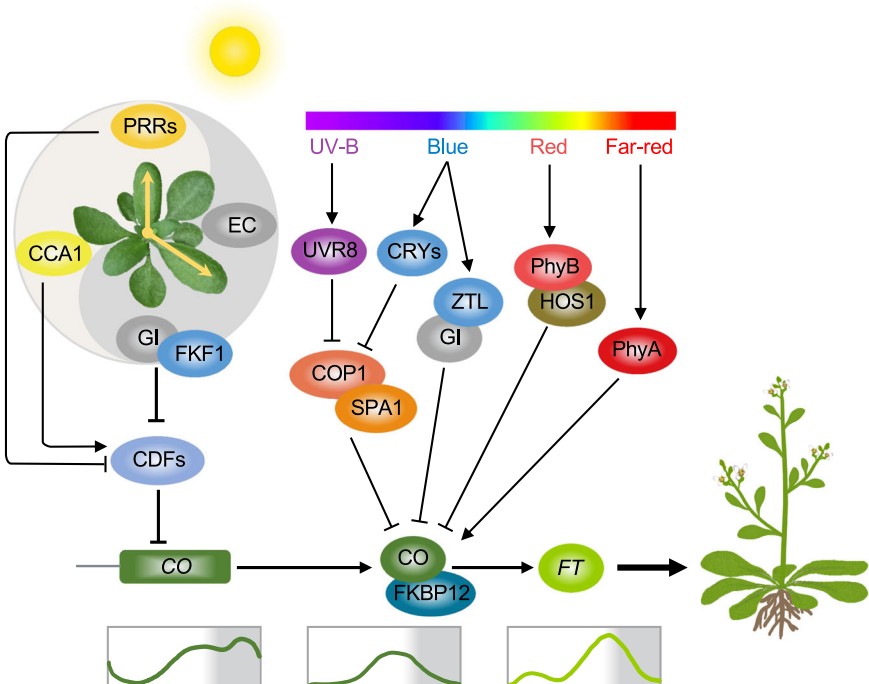

**Fig. 2 | Photoperiodic flowering pathway in Arabidopsis.** *CO* and *FT* are key regulators in the photoperiodic flowering pathway. CO is tightly regulated by the circadian clock and external light signals, and it activates *FT* expression to initiate flowering. The interaction between FKBP12 and CO protein prevents CO degradation and modulates its nuclear localization. In the early morning, CDF transcription factors that are regulated by the circadian clock accumulate at high levels and directly repress *CO* expression, thereby delaying flowering. GI is another key player in regulating circadian outputs. Under long-day conditions, high levels of GI work together with the blue-light receptor FKF1 to facilitate the degradation of the flowering repressor CDFs, promoting the initiation of flowering. Conversely, the core clock component CCA1 promotes the expression of *CDFs* by directly binding to their promoters, which enhances the inhibitory effect of CDFs on *CO* expression, leading to delayed flowering. Additionally, PRRs can directly or indirectly regulate *CDF* expression, affecting flowering. The light-sensitive proteins CRYs, UVR8, PhyB, and PhyA are capable of sensing blue, UV-B, red, and far-red light, respectively, and modulating the activity of CO protein to regulate *FT* expression and flowering. The line graphs at the bottom panel indicate mRNA oscillation of *CO*, *CO/FKBP12*, and *FT*, respectively. Arrows, lines, and shades are the same as in Fig. 1.

adjusting the rhythmic expression of the clock component *GI* and the F-box protein *FKF1*, and this leads to the appropriate timing of *CO* expression and photoperiodic flowering[35].

The photoperiodic flowering pathway is regulated by the interaction of various proteins. GI together with the blue-light receptors ZTL, LKP2, and FKF1 regulates the degradation of the flowering repressor CDFs to control the clock-mediated photoperiodic flowering pathway[57]. Under long-day conditions, the timing of *GI* and *FKF1* expression is synchronized, peaking in the late afternoon, and the blue-light absorption enables FKF1 to form a sufficient protein complex with GI to target CDF proteins for degradation, which alleviates the transcriptional repression of the *CO* gene and promotes flowering[50]. In contrast, diurnal expression of FKF1 and GI proteins is out of phase during short days, leading to low levels of FKF1-GI complex and *CO* expression under light, and consequently, the flowering is delayed. A computational model predicts that FKF1 may control *FT* expression directly in addition to its role in *CO* transcriptional activation[58]. *ZTL* and *LKP2* also play important roles in regulation of flowering time and the circadian clock by ubiquitin-mediated degradation of clock components. Introducing both *ztl* and *lkp2* mutations into the *fkf1* mutant further enhances the late-flowering phenotype of the *fkf1* mutant and reduces the *CO* expression level, due to the increased abundance of CDF2 protein[59]. In contrast to *FKF1*, overexpression of *ZTL* or *LKP2* leads to an unexpected late-flowering phenotype accompanied with low levels of *CO* expression, similar to the *ztl fkf1 lkp2* triple mutant and the *gi* mutant[60]. One explanation for this late-flowering phenotype of these overexpressing lines is that a high level of the ZTL/LKP2 sequester FKF1 in the cytosol by forming the ZTL/LKP2-FKF1 complex. This reduces the amount of FKF1-GI-CDF1 complex in the nucleus, causing repression of *CO* expression[61]. Another possibility is that a

high level of ZTL/LKP2 destabilizes core clock components TOC1 and PRR5, which interact with and stabilize CO protein to promote flowering in response to the day length[40,41,55].

## Light controls CO protein stability in Arabidopsis

Transcript levels of *CO*, *CO/FKBP12*, and *FT*, respectively, are diurnally oscillated (Fig. 2, bottom panel). At the post-transcriptional level, the stability of CO protein is compromised by the action of post-translational modification. The degradation of CO protein is mainly controlled by two RING-finger E3 ubiquitin ligases, HIGH EXPRESSION OF OSMOTICALLY RESPONSIVE GENE1 (HOS1)[62] and CONSTITUTIVE PHOTOMORPHOGENIC 1 (COP1)[63], which act in a light-dependent manner. HOS1 directly targets to CO protein to promote its degradation in the early morning, and mutation in *HOS1* causes extreme early flowering in both long- and short-day conditions[62]. However, COP1 promotes CO degradation during night, leading to suppression of the floral integrator *FT* and delayed flowering[64], and SUPPRESSOR OF PHYA (SPA1) can enhance the E3 ubiquitin ligase activity of COP1[65]. In contrast, the small immunophilin FK506 BINDING PROTEIN 12 KD (FKBP12) physically interacts with CO to prevent its degradation by COP1, and this interaction can also modulate its phosphorylation and nuclear localization, enabling it to trigger *FT* expression and flowering[66].

Photoreceptors also play critical roles in regulating CO stability and act antagonistically to generate daily rhythms in CO abundance[14]. The blue-light photoreceptors cryptochrome1 (CRY1) and CRY2 physically interact with SPA1 to suppress the activity of COP1/SPA1 ubiquitin ligase and stabilize CO protein in a blue-light-dependent manner[67]. The red/far-red responsive phytochromes (PhyA–PhyE in Arabidopsis) also

control flowering time[68]. Among the five phytochromes, PhyA and PhyB play the most predominant function in CO protein stability[8,69]. Under the far-red light, PhyA stabilizes CO protein and promotes flowering by inhibiting the COP1-SPA1 complex, while red light-activated PhyB interacts with HOS1 to promote CO protein degradation[14,62]. The light-dependent control of CO protein stability is a complex and tightly regulated process that controls photoperiodic flowering. The knowledge gained from Arabidopsis research has provided fundamental insights into the mechanisms controlling photoperiodic flowering in other plant species including agricultural crops.

**Photoperiodic flowering mechanism in rice**

The function of core circadian regulators is conserved among Arabidopsis and crops. For example, the function of *CCA1*, *LHY*, and *TOC1* is conserved in rice[70,71], maize[72], and duckweed[73]. Overexpressing maize *ZmCCA1b* and rice *OsPRR* homologs, respectively, can complement mutant phenotypes in Arabidopsis[71,72]. In rice, *OsCCA1* upregulates expression of strigolactone receptor and signaling and responsive genes to repress tiller-bud outgrowth[70]. Down-regulating and overexpressing *OsCCA1* increases and reduces tiller numbers, respectively, while altering *OsPRR1* expression leads to the opposite effects. Moreover, both exogenous and endogenous sugars negatively regulate expression of *OsCCA1*, which in turn mediates expression of strigolactone receptor and signaling pathway genes to control tillering, flowering, and panicle development.

Rice is a facultative short-day plant; flowering is accelerated under short-day conditions and repressed by long-day conditions (Fig. 3). The floral transition in rice is dependent on the transcription of two florigen genes *Heading date3a* (*Hd3a*) and *Rice FT1* (*RFT1*)[74–76], which are mainly controlled by two important transcription factors *Heading date1* (*Hd1*)[77], an ortholog of the Arabidopsis *CO*, and *Early heading date1* (*Ehd1*)[78] that is unique in rice. Similar to Arabidopsis *FT*, both *Hd3a* and *RFT1* are expressed in the vascular tissue of leaves and move to the shoot apical meristem, where they enhance the expression of floral meristem identity genes, such as *OsMADS14* and *OsMADS15*, and trigger flowering[75,76]. *Hd3a* and *RFT1* are essential for promoting rice flowering under short-day conditions, while *RFT1* functions as a floral activator under long-day conditions. At least two distinct pathways regulate rice floral transition, the conserved *OsGI-Hd1-Hd3a* pathway and a unique *Grain number, Plant Height, and Heading date1* (*Ghd7*)-*Ehd1-Hd3a/RFT1* pathway (Fig. 3), which are both regulated by the circadian clock and light perception[8].

Unlike *CO* in Arabidopsis, the *Hd1* gene in rice is regulated by the circadian component *OsGI* and displays bifunctional responses to day length: promoting flowering by activating the expression of *Hd3a* under short-day conditions, but repressing *Hd3a* expression to delay flowering under long-day conditions[77,79]. This photoperiod-dependent conversion of *Hd1* function from activation to repression is modulated by phytochromes and circadian clock. *Photoperiodic Sensitivity5* (*SE5*) encodes a putative heme oxygenase involved in the biosynthesis of the phytochrome chromophore. In the *se5* mutant, *Hd3a* expression is strongly induced by Hd1 under long-day conditions, resulting in early flowering phenotype similar to phytochrome-deficient mutants, such as *OsphyB* and *OsphyC*[80,81]. Furthermore, overexpression of *Hd1* can repress *Hd3a* expression and delay flowering under short-day conditions, and this effect requires functional phytochromes[82]. Therefore, phytochrome signals are essential for the conversion of Hd1 from an activator to a repressor of *Hd3a* transcription.

The rice-specific regulator *Ghd7* encoding a CCT (CO, CO-LIKE, and TIMING OF CAB1) domain transcriptional regulator was identified as a major target of phytochrome signals in flowering control and acts as a repressor of *Ehd1*[78,83,84]. *Ehd1* is an important B-type response regulator that activates expression of *Hd3a* and *RFT1*, which are responsible for the transition from vegetative to reproductive growth in rice[78,84]. *Ghd7* acting as a repressor of flowering is acutely induced by red light in morning through the PhyA homodimer or the PhyBPhyC heterodimer, and the repressor activity of Ghd7 is modulated post-transcriptionally by PhyB. Specifically,

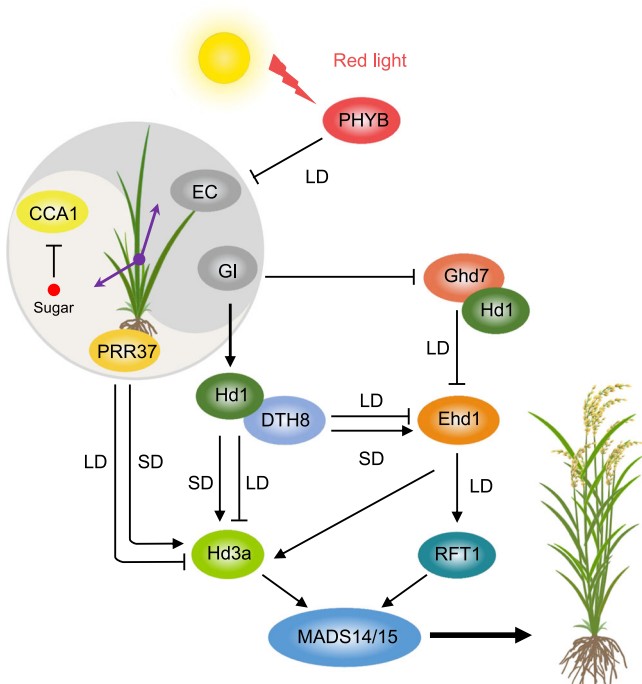

**Fig. 3 | Circadian and photoperiodic regulation of flowering pathways in rice.** Rice has evolved two distinct photoperiodic flowering pathways to adapt varying short (SD) and long (LD) day lengths. The evolutionarily conserved *Hd1-Hd3a* pathway is activated under SD conditions and the uniquely evolved *Ghd7-Ehd1-Hd3a/RFT1* pathway is activated under LD conditions, and they are regulated by both the circadian clock and light signaling. Hd1 plays a crucial role in regulating rice flowering time by interacting with the NF-YB transcription factor DTH8 in response to day length changes. Under SD conditions, Hd1 promotes the expression of *Hd3a* and *Ehd1* to activate floral meristem identity genes. However, under LD conditions, Hd1 suppresses the expression of *Hd3a* and *Ehd1* to prevent premature flowering, contributing to the photoperiodic adaptability of rice. Hd1 also interacts with the flowering repressor Ghd7 to inhibit *Ehd1* expression, thereby delaying flowering under LD conditions. OsGI triggers the initiation of the *Hd1-Hd3a* pathway by activating *Hd1* expression and can enhance *Hd3a* and *Ehd1* expression by inhibiting Ghd7, leading to flowering. The clock gene *OsPRR37* also regulates photoperiodic flowering by activating *Hd3a* expression under SD conditions and suppressing it under LD conditions, allowing rice plants to fine-tune flowering timing for optimal reproductive success under changing environmental conditions. Sugar suppresses expression of *OsCCA1*, which in turn controls strigolactone pathway genes to regulate tiller buds and flowering. *MADS14/15* are floral identity genes. Arrows, lines, and shades are the same as in Fig. 1.

when PhyB is activated by red light, it undergoes a conformational change that allows it to interact with Ghd7 protein, and this interaction leads to the ubiquitination and subsequent degradation of Ghd7 through the 26 S proteasome pathway, which relieves its repressor activity and promotes flowering[85]. *Days to heading8* (*DTH8*) encodes a putative HAP3 subunit of NF-YB transcription factor that is capable of binding to *Hd3a* promoter to enhance H3K27 trimethylation and repress *Hd3a* expression[86,87]. Under long-day conditions, both Ghd7 and DTH8 can interact with Hd1 to form a strong repressor complex that inhibits *Ehd1* expression, and this effect of the Hd1-Ghd7-DTH8 complex on *Ehd1* expression can also partly explain the photoperiod-dependent conversion of Hd1 from activation to repression[86,88,89].

In sorghum, SbPRR37 activates expression of several downstream genes that repress flowering in long days[90]. *SbPRR37* expression is dependent on light and regulated by the circadian clock. In short days, *SbPRR37* is not expressed during the evening phase, allowing sorghum to flower. Similarly, clock components such as *OsPRR37/DTH7* and the EC

components, also regulate rice flowering by enhancing photoperiod sensitivity. *OsPRR37* is predicted to act downstream of the OsPhyB to switch the genetic effects of Hd1 on *Hd3a* expression and delay flowering in long-day conditions through the formation of a transcriptional repressor complex[91–93].

The EC is a crucial flowering repressor[33]. *OsELF3-1* and *OsELF3-2*, two rice *ELF3* paralogs, physically interact with OsELF4a in the nucleus, whilst OsELF3-1 shows a stronger interaction with OsLUX compared to OsELF3-2[94]. Recent studies have shown that functional OsELF3 proteins accumulate during the evening in short-day conditions, forming the EC to induce flowering by inhibiting the expression of floral repressors *OsPRR37* and *Ghd7*[93]. However, in long-day conditions, OsELF3 protein levels are low and insufficient to suppress floral repressors, resulting in delayed flowering[95]. Taken together, these data indicate that rice possesses several complex genetic pathways to affect flowering time and photoperiod sensitivity.

## Photoperiodic flowering mechanism in soybean

Soybean is a sensitive short-day plant, and this sensitivity limits the regional adaptation and crop yield. A series of genes have been identified to play a role in fine-tuning soybean flowering and maturity time and improve the regional adaptation (Fig. 4). The genome of soybean, being a paleopolyploid[96], has at least 12 *FT* homologs. Among these, *GmFT2a* and

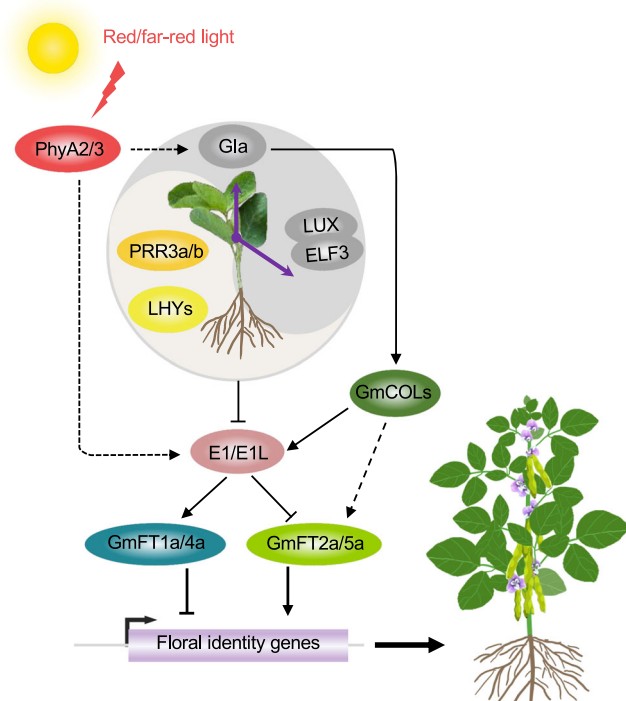

**Fig. 4 | Genetic control of photoperiodic flowering in soybean.** Soybean has multiple *FT*-like genes, and GmFT2a and GmFT5a promote flower formation by enhancing the expression of flower meristem identity genes, including *GmSOC1*, *GmAP1*, and *GmLFY*. Conversely, GmFT1a and GmFT4a inhibit flower formation by repressing these genes. The central regulator E1 inhibits soybean flowering by downregulating expression of *GmFT2a* and *GmFT5a* and upregulating expression of *GmFT1a* and *GmFT4a*. Under LD conditions, *E1* expression is strongly induced by the red/far-red light receptors GmPhyA2 and GmPhyA3, leading to delayed flowering. These photoreceptors can also transmit light signals to circadian clock factor homologs such as GmGIa, GmPRR3a/b, GmELF3, and GmLUX. These factors mediate expression of *GmLHYs*, leading to distinct expression rhythms under SD and LD conditions, affecting flowering time in soybean plants. *GmCOLs* and *GmFT2a/5a* may form *CO-FT* regulons and play an important role in photoperiodic flowering in soybean. These models are based on genetic and gene expression results and may be refined by biochemical studies. Arrows, lines, and shades are the same as in Fig. 1, while dashed arrows indicate predicted interactions.

*GmFT5a* are considered to be major functional *FT* orthologues, as they are highly induced under short-day conditions[97]. These genes promote flowering in response to short-day conditions, which is important for soybean adaptation to short-day regions. On the contrary, *GmFT1a* and *GmFT4a* appear to be inhibitors of flowering, and they are highly induced under long-day conditions[98,99]. These genes prevent premature flowering in response to long-day conditions, which is important for soybean adaptation to long-day regions. The remaining 8 *FT* homologs are nonfunctional or expressed at low levels in soybean leaves under inductive short-day conditions[100]. Flowering QTLs are associated with circadian clock regulators, *GI* in soybean[101], and *CO* in sorghum[102]. The soybean genome has 28 *CONSTANS-like* genes (*GmCOLs*), and some of these genes, including *GmCOL1a* and *GmCOL1b*, are induced under long-day conditions and can activate the expression of florigen genes *GmFT2a* and *GmFT5a*, promoting soybean flowering[103]. Other *GmCOLs* have different expression patterns and functions in regulating soybean photoperiodic flowering[104].

The flowering repressor gene *E1* encodes a legume-specific transcription factor that induces *GmFT1a* and *GmFT4a* expression but represses *GmFT2a* and *GmFT5a* expression. This gene is directly repressed by four soybean *CCA1/LHY* genes (*GmLHY1a*, *GmLHY1b*, *GmLHY2a* and *GmLHY2b*) at the transcriptional levels[98,99,105]. Indeed, the soybean quadruple knockout mutant *lhy1a lhy1b lhy2a lhy2b* generated by CRISPR-Cas9 displayed late flowering and high *E1* transcript levels in long days[105]. Meanwhile, *GmPRR3a* and *GmPRR3b*, orthologs of Arabidopsis *PRR3*, induce *E1* expression to delay photoperiodic flowering through downregulation of *GmLHY* genes[105,106]. This pathway provides an additional layer of regulation for soybean flowering. The *E2* locus in soybean encodes a homolog (*GmGIa*) of the Arabidopsis *GI*, which acts upstream of *CO* and *FT* in the photoperiodic flowering pathway. Like the role of *GI* in Arabidopsis, *GmGIa* plays a crucial role in integrating light and circadian signals in response to photoperiodic changes, ultimately leading to the transition from vegetative growth to reproductive development[101]. GmGIb and GmGIc are two homologs of GmGIa and can interact with two Arabidopsis FKF1 orthologs (GmFKF1 and GmFKF2) and one Arabidopsis CDF1 ortholog (GmCDF1) proteins, but GmGIa cannot interact with these three proteins, suggesting that GmGIa may play a unique role in regulating soybean flowering[107]. GmELF3, an ortholog of clock component ELF3, physically interacts with GmLUX2 to directly repress *E1* expression and promotes soybean flowering[108].

In addition to these genes, two maturity loci, *E3* and *E4*, encode the phytochrome A proteins GmPhyA3 and GmPhyA2, respectively. These proteins induce expression of *E1* and which, in turn, suppresses expression of *GmFT2a* and *GmFT5a*, resulting in delayed flowering under both natural day length and artificially long-day conditions[109,110]. The PhyA-regulated E1-GmFT pathway is a key determinant for soybean adaption to different latitude environments. The blue-light photoreceptor protein cryptochromes also play an essential role in regulating circadian rhythm and flowering time. Soybean contains two cryptochrome genes, GmCRY1a and GmCRY2a. However, only GmCRY1a has been found to strongly promote floral initiation in soybean, and the circadian rhythm of GmCRY1a protein has different phase characteristics in different photoperiods, which suggests that GmCRY1a plays a more predominant role in regulating flowering time in response to changes in day length[111]. The role of *GmCRY2a* in soybean flowering remains unclear and to be further investigated to elucidate its function.

## Epistasis and epigenetic regulation of photoperiodic flowering in Arabidopsis and crops

Epigenetic regulation of circadian and flowering-time pathway genes plays a broader role in determining photoperiodic flowering in plants. Epigenetics refers to heritable changes in gene expression that do not involve alterations to the DNA sequence. Epigenetic mechanisms, involving DNA methylation, chromatin modifications, long-noncoding RNAs, and small interferring RNAs, have been shown to regulate expression of key photoperiodic flowering pathway genes and flowering time.

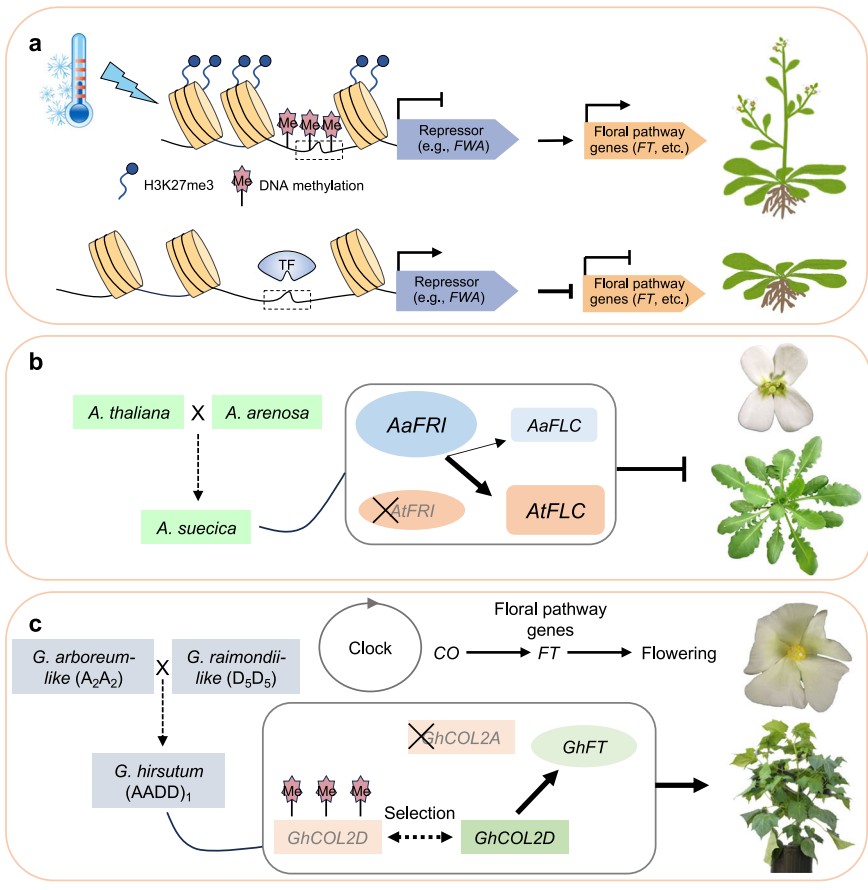

**Fig. 5 | Epigenetic and epistatic regulation of flowering time in Arabidopsis and polyploid plants. a** FWA and FLC proteins inhibit flowering pathway genes by binding to their promoters or proteins. In the wild-type, *FWA* locus was methylated to suppress its activity. In the *fwa* mutant or epiallele induced by *DDM1*, *FWA* is demethylated to activate its expression, which represses floral pathway genes such as *FT* and *FLC* to inhibit flowering. Vernalization can induce long-noncoding RNAs like COOLAIR or COLDAIR in the first intron of *FLC* locus and H3K27me3 through polycomb-group2 (PRC2) protein complex, which coincides with repression of *FLC* to promote flowering. **b** In *Arabidopsis suecica* allotetraploids that were formed from *A. thaliana* and *A. arenosa*, AtFRI is silenced, while AaFRI is expressed. AaFRI *trans*-actives AtFLC expression, consistent with a wide range of flowering phenotypes in resynthesized and natural *A. suecica*. **c** In allotetraploid Upland cotton (*Gossypium hirsutum*, $A_2A_2D_5D_5$), *CONSTANS-LIKE2* (*COL2*) has two homoeologs *GhCOL2A* and *GhCOL2D*. GhCOL2A promoter is heavily methylated and silenced, while *GhCOLD2* is a reversible epiallele, which is methylated and repressed in wild relatives of *G. hirsutum* but demethylated and expressed in cultivated cotton to promote *GhFT* expression and photoperiodic flowering. This may contribute to worldwide cultivation of Upland cotton. Photos are not in scale.

One of the best-studied epigenetic flowering events is known as vernalization, which refers to the induction of flowering after prolonged exposure to cold temperatures in winter, mediated by changes in the chromatin status of *FLC*[112]. *COLD INDUCED LONG ANTISENSE INTRAGENIC RNA* (*COOLAIR*)[113] and *COLD ASSISTED INTRONIC NONCODING RNA* (*COLDAIR*)[114] originated from the first intron and 3' end of *FLC*, respectively. They have been proposed to facilitate *FLC* silencing by removing H3K36me3 from chromatin during vernalization, particularly under low-temperature conditions[113,114]. However, recent research suggests that the *COOLAIR* is not required for vernalization. Support for this notion comes from the normal vernalization response observed in the CRT/DRE-binding factors (CBFs) *cbfs* mutants with reduced levels of *COOLAIR* induction; moreover, both *cbfs* and $FLC_{\Delta COOLAIR}$ mutants show a normal vernalization response despite their inability to activate *COOLAIR* expression during cold[115]. This work also highlights the importance of CBFs in initiating *COOLAIR* expression during the early stage of vernalization. CBFs bind to CRT/DREs at the 3'-end of *FLC* and increase progressively during vernalization, promoting *COOLAIR* expression. As vernalization progresses, *FLC* chromatin shifts to an inactive state, prompting CBF proteins to detach from the CRT/DREs in the *COOLAIR* promoter, thereby diminishing *COOLAIR* levels.

In Arabidopsis, *FWA* is an epiallele and encodes a flowering suppressor capable of specifically inhibiting the function of FT by directly binding to the FT protein[17,116] (Fig. 5a). In the wild type, two repeats in the promoter region of *FWA* is heavily methylated and its expression is repressed, while the *FWA* promoter is demethylated and *FWA* is expressed in the *fwa* mutant[17]. Removal of DNA methylation in the *FWA* promoter can lead to activation of *FWA* and late flowering, as observed in multiple late-flowering mutants that were found in the *decrease in DNA methylation1* (*ddm1*) genetic background[16]. These epialleles likely result from the plant adaptation in response to changing environments.

Epialleles or epigenes can also result from cross-fertilization such as imprinting[117,118] and polyploidy[25,26]. In Arabidopsis intraspecific hybrids[119,120] and allotetraploids[121], expression waveforms of circadian clock genes such as *CCA1* and *LHY* are altered by epigenetic and chromatin modifications to increase photosynthesis, chlorophyll biosynthesis, and starch metabolism[121]. The more starch accumulates during the day, the more it can be degraded at night to promote growth vigor, in a widespread phenomenon known as hybrid vigor or heterosis[24,27]. Stress-responsive gene expression is gated by the circadian clock[122]. Altered circadian gene expression in the hybrids also regulates expression of abiotic and biotic stress-responsive genes as a trade-off to balance the energy used for growth and defense[119], as well as mediates ethylene biosynthesis that in turn regulates growth vigor[123]. Mechanistically, *CCA1* expression changes are related to parent-of-origin effect on DNA methylation[120], suggesting an epigenetic cause. Further analysis showed that circadian clock genes mediate diurnal regulation of histone H3K4 methylation reader and eraser genes, which in turn regulate rhythmic histone modification dynamics for the clock and its output genes[124]. This reciprocal regulatory module between chromatin modifiers and circadian clock oscillators orchestrates diurnal gene expression that governs plant growth and development.

All flowering plants are polyploids or of polyploid origin[125,126], which result in genetic and epigenetic changes[25,26]. Duplication of flowering pathway genes in polyploids may increase the range of flowering time variation or induce other changes including epigenetic modifications and epistasis or *trans*-acting effects[25,127,128]. Natural variation of flowering time in *Arabidopsis suecica* allotetraploids is largely controlled by two epistatically acting loci, namely *FRIGIDA* (*FRI*) and *FLC*[129,130]. *FRI* upregulates *FLC* expression that represses flowering[131]. In Arabidopsis allotetraploids that are formed by pollinating *A. arenosa* with *A. thaliana*[132,133], there are two sets of *FLC* and *FRI* genes and they flower late[128]. Inhibition of early flowering is caused by upregulation of

*A. thaliana FLC* (*AtFLC*) that is *trans*-activated by *A. arenosa FRI* (*AaFRI*) (Fig. 5b). Two duplicate *FLCs* (*AaFLC1* and *AaFLC2*) originating from *A. arenosa* are expressed in some allotetraploids but silenced in other lines. The expression variation of *FLCs* in the allotetraploids is associated with deletions in the promoter regions and first introns of *A. arenosa FLCs*. The strong *AtFLC* and *AaFLC* loci are maintained in natural Arabidopsis allotetraploids, leading to extremely late flowering[128]. Furthermore, *FLC* expression correlates positively with histone H3K4me2 and H3K9ac marks and negatively with the H3K9me2 mark. This combination of epistasis between loci and chromatin regulation within a locus may provide a complexity of flowering time variation in polyploid plants in response to environmental cues and adaptive niches.

Cotton allotetraploids were formed ~1–1.5 million years ago (Mya)[134] (Fig. 5c), followed by natural diversification and crop domestication[135]. Polyploidization between an A-genome African species (*Gossypium arboreum*-like) and a D-genome American species (*G. raimondii*-like) in the New World created a new allotetraploid or amphidiploid (AD-genome) cotton clade[134], which has diversified into five polyploid lineages, *Gossypium hirsutum* (Gh) (AD)$_1$, *G. barbadense* (Gb) (AD)$_2$, *G. tomentosum* (AD)$_3$, *G. mustelinum* (AD)$_4$, and *G. darwinii* (AD)$_5$. *Gossypium ekmanianum* and *G. stephensii* are recently characterized and closely related to *G. hirsutum*[136]. Gh and Gb were separately domesticated from perennial shrubs to become annualized Upland and Pima cottons[135].

Interspecific hybridization induces genome shock, including non-additive gene expression and epigenetic changes in polyploid plants and crops[133,137]. The epigenetic changes produce epigenes or epialleles, which are selected during evolution and domestication. In allotetraploid cotton, over 500 epigenes have been identified and maintained over 600,000 years of genomic diversification, some of which are predicted to originate during polyploidization over 1–1.5 million years ago[22]. Many of these epigenes are related to domestication traits including flowering time, stress response, seed development, and seed dormancy.

Cotton, being originated in tropical and subtropical areas, flowers in short-day conditions[138]. Loss of photoperiod sensitivity is a major "domestication syndrome" trait[139] of Upland or American cotton (*G. hirsutum* L.) and Pima or Egyptian cotton (*G. barbadense* L.) that accounts for >95% and ~5% of annual cotton crop worldwide, respectively[140]. This process is associated with cotton *CO*-liked (*COL*) genes, which are Arabidopsis *CO* homolog[141]. Among 23 *COLs* identified in cotton[142], eight *G. hirsutum COLs* (*GhCOLs*) are in the same subgroup. Among them, only *GhCOL2* exhibited similar expression rhythms with *GhFT*, indicating that *GhCOL2* is a major regulator of *GhFT*.

Allotetraploid cotton has two *COL2* homoeologs, *GhCOL2A* and *GhCOL2D* (Fig. 5c). *COL2D* is heavily methylated and silenced in *G. raimondii* that is photoperiod sensitive, while *COL2A* is hypomethylated and highly expressed in cultivated *G. arboreum*, which is photoperiod insensitive[138]. In allotetraploid cotton, the *COL2A* homoeolog was hypermethylated and repressed in cultivated Upland and Pima cottons, while the *COL2D* homoeolog was highly expressed. This suggests that *COL2A* in the allotetraploid cottons is likely silenced after polyploid formation since it is expressed in the extant *G. arboreum* species. The high-level expression of *COL2D* is likely associated with positive selection and loss of DNA methylation of *COL2D* during domestication of Upland and Pima cottons[142], which could lead to the loss of photoperiod sensitivity for global cotton cultivation. Indeed, methylation levels of *COL2D* are lower and their expression levels are higher in cultivated and photoperiod-insensitive *G. hirsutum* and *G. barbadense* than in their photoperiod-sensitive wild relatives. Removal of DNA methylation in the wild *G. hirsutum* (TX2095) seedlings using 5'-aza-2'-deoxycytidine (5-aza-dC), a chemical inhibitor for DNA methylation[143], activates *COL2D* expression. Moreover, using virus induced gene silencing (VIGS)[144], *GhCOL2* is down-regulated, which is consistent with the repression of *GhFT*. Down-regulating *COL2* and *GhFT* expression has delayed flowering time for 9 days, compared to the control plants and wild cottons that do not flower in the LD conditions[138]. This example has demonstrated

important roles of epigenes and epialleles in photoperiodic flowering during natural selection and crop domestication. These epigenes and epialleles can be candidate targets for gene-editing and biological breeding to improve crop production and resilience.

## Concluding remarks

Growth, development, and reproduction during appropriate times of the year is essential for plants to adapt to seasonal changes. The photoperiodic response to flowering is a key mechanism for plants to optimize utilization of natural resources, adapt crop varieties in regional environments, and significantly impact crop yield stability and potential. Insights gained from studying the interplay between light signaling, the circadian clock, and photoperiodism in the model plant Arabidopsis have facilitated the discovery of complex molecular circuitry networks underlying photoperiodic flowering pathways in both short- and long-day crops, many of which are polyploids and have genomic redundancy that is subject to epigenetic modifications. Indeed, some key regulators, including circadian clock genes, are epigenetically regulated, presumably resulting from natural selection, crop domestication, and/or modern breeding.

Genome sequencing and genome-wide association studies in crop plants have supported that allele variants of several key circadian genes have been selected by breeding and under agricultural culture of adapting crop flowering time to certain seasons. While most current studies are focused on light and circadian control of photoperiodic flowering, temperature changes can drastically affect plant flowering and have been grossly under-investigated. Further investigations are needed to maintain agricultural and ecological sustainability in response to changes in extreme weather and climate such as drought, heat, and flood.

There are good examples of the genes in the circadian clock and photoperiodic flowering pathways that are associated with agronomic traits in crops. Rice *Hd1*[77], an ortholog of the Arabidopsis *CO*, controls expression of two florigen genes *Hd3a* and *RFT1*[74–76] during the floral transition. *ZmCCT9* in maize is diurnally regulated and negatively regulates the expression of the florigen *ZCN8* through a CACTA-like TE insertion in *ZmCCT* promoter region[145], resulting in reduction of photoperiod sensitivity, thus accelerating maize spread to long-day environments[146]. The circadian clock of cultivated tomatoes has slowed during domestication. *EMPFINDLICHER IM DUNKELROTEN LICHT1* (EID1) is an F-box protein that targets phytochrome A for degradation in Arabidopsis[147]. The *EID1* allele in cultivated tomatoes is under selection sweep and enhances plant performance under long-day photoperiods presumably resulting from moving away from the equator[148]. Fruit yield in hybrid tomato is related to *SINGLE FLOWER TRUSS* (*SFT*)[149], a homolog of *FT* in Arabidopsis. Moreover, epigenetic alteration of circadian clock genes including *CCA1* and *LHY* in plant hybrids are related to growth vigor in Arabidopsis[119–121] and maize[72,150]. In cotton, *GhCOL2* is an epiallele, which may be selected during domestication to increase its expression and thereby reduce photoperiod sensitivity, helping spread worldwide cotton cultivation[22]. These examples will help us design strategies to optimize circadian input and output signals and improve flowering time and crop yield using genome-editing tools. Likewise, discovery and utilization of new epigenes and epialleles in combination of targeted-gene editing will improve flowering time and crop yield and resilience.

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

## Acknowledgements
This work was supported in part by National Natural Science Foundation of China (32200264, F.W.), National Natural Science Foundation of Shandong (ZR2022QC050, F.W.), Taishan Scholar Project of Shandong Province of China (tsqn202211101, T.H.), National Institute of General Medical Sciences (GM109076, Z.J.C.), National Science Foundation (ISO1238048 and IOS1739092, Z.J.C.), a Stengl-Wyer Endowment Research Grant (2022–2024, Z.J.C.), and the Winkler Fellowship (2024–2025, Z.J.C.).

## Author contributions
F.W. and Z.J.C. conceived the article and wrote the manuscript. F.W., T.H., and Z.J.C. made figures. F.W., T.H., and Z.J.C. revised and approved the article.

## Competing interests
The authors declare no competing interests.
