## [Peer review file · Communications Biology]

Reviewers' comments:

Reviewer #1 (Remarks to the Author):

The review by Wang et al. provides a comprehensive description of the genetic pathways underlying photoperiodism and circadian responses the vegetative to reproductive transition in different plant species. This is an important topic that influences crop adaptation to different environments and has been the subject of much genetic research which is described in this review.

The manuscript contains a large number of references and I have no major concerns with the validity of the statements made about gene function and the interactions between various components of these pathways.

However, I found the structure of the paper and the areas of focus to be unusual and made the paper, for me, difficult to follow. In sections, the manuscript read as a list of statements about gene functions and lacked a strong narrative that tied all this information together. This may have been the intention of the authors and for their target audience, just that I found the manuscript quite challenging to follow.

Photoperiodism is such a large area of research that I think a review would benefit from greater focus and a more logical structure. In addition to the lack of a single narrative flow, basic concepts of photoperiodism (e.g. short day and long day requirements to flower, circadian gating) are left undescribed even at a high level, in place of extremely detailed descriptions of the interactions between a great number of genes. Information is grouped unusually, with unrelated topics included in the same sections and other sections that do not have an obvious link to photoperiodism (such as cotton evolution) which might be better described in a different manuscript.

Some specific comments on the structure:

- Photoperiodism and circadian regulation affects many developmental processes, but it should be clear in the introduction, abstract and title that the focus of the review is on the vegetative to reproductive transition. The current title is very broad and the introduction section mentions different biological processes that are not relevant to the current study. A greater focus on the vegetative to reproductive transition would clarify for the reader what the focus of the paper is.
- The introduction section is potentially unnecessary and describes a lot of information on genes and their potential roles that are later repeated in specific sections.
- Many concepts are not described adequately. It is not stated that Arabidopsis is a long-day plant and that these pathways have adapted to different photoperiods to promote flowering. The concept of circadian gating is not described, nor the temperature-mediated changes in the clock.
- There is a section combining epistasis and epigenetics, two different topics, that are covered very briefly with few examples and do not seem to be especially important for photoperiodism (These are facets across gene regulation). The inclusion of epistasis in this section is particularly unusual. The section then moves to description of polyploidization in cotton and a description of genes in this species. I understand this is an area of research expertise for the authors, but the topic here does not have special interest for photoperiodism and should probably be left out.
- The concluding remarks section mentions that an understanding of these pathways can lead to engineering yield improvement. It would be good to include some examples of crops for which this has been achieved to demonstrate the real-world outcomes of this research area.

Overall, I feel that the manuscript could be improved by clearly defining the topic and target audience, then rewriting to focus on these topics so they are most clearly understood, with a grounding in some of the underlying concepts. This will help readers to understand the relevance of each gene pathway and the broader context of how they fit in to plant biology.

Minor comments

Line 41: Replace "Despite" with "although".

Line 42: "key regulators of floral transition are highly conserved". This is poorly defined. I am familiar with work from wheat that shows the CONSTANS proteins, for example, have a less important role in defining flowering time in this species. I believe this sentence should either be modified or removed.

Line 84 – Sentence is incomplete, ends on "constitute".

Lines 113-116: These sentences seem out of place since they describe the conservation of

regulators in different species. Was this the topic of the introduction?

Line 203: "switch on flowering" could be defined in more technical terms.

Lines 238 – 242: The genes mentioned here are from Soybean. Why is this information not grouped together in the section describing responses in soybean?

Line 246: The evening complex in *Arabidopsis* is introduced mid-paragraph.

Figure 2: What are the line graphs at the bottom of the figure?

Reviewer #2 (Remarks to the Author):

This submission by Wang et al. provides a timely and comprehensive overview of the molecular mechanisms underlying photoperiodic regulation of flowering times in the model plant *Arabidopsis thaliana* and several economically-important crops. Overall the manuscript is well written. Below are suggestions for improvement.

1. Title: it would be better to remove the word of crops because crops are plants.
2. Lines 43-58: As these two paragraphs describe the mechanisms in *Arabidopsis thaliana*, not in other plants, please add the species name somewhere in the first paragraph.
3. Line 84:constitute?
4. Lines 312-317: the function of COOLAIR in vernalization-mediated *FLC* repression has been questioned (e.g. DOI: 10.7554/eLife.84594 and other publications). Please rephrase this paragraph.
5. A few typos. Line 91: actives to activates. Line 114: ...conserved IN duckweed... Line 135: diel to diurnal? Line 311: ...cold IN winter... Line 203: please italicize *OsMADS14* and *OsMADS15*.
6. Lines 318-320: these two sentences are not consistent.

Point-by-point response

Reviewer #1 (Remarks to the Author):

The review by Wang et al. provides a comprehensive description of the genetic pathways underlying photoperiodism and circadian responses the vegetative to reproductive transition in different plant species. This is an important topic that influences crop adaptation to different environments and has been the subject of much genetic research which is described in this review.

The manuscript contains a large number of references and I have no major concerns with the validity of the statements made about gene function and the interactions between various components of these pathways.

However, I found the structure of the paper and the areas of focus to be unusual and made the paper, for me, difficult to follow. In sections, the manuscript read as a list of statements about gene functions and lacked a strong narrative that tied all this information together. This may have been the intention of the authors and for their target audience, just that I found the manuscript quite challenging to follow.

Photoperiodism is such a large area of research that I think a review would benefit from greater focus and a more logical structure. In addition to the lack of a single narrative flow, basic concepts of photoperiodism (e.g. short day and long day requirements to flower, circadian gating) are left undescribed even at a high level, in place of extremely detailed descriptions of the interactions between a great number of genes. Information is grouped unusually, with unrelated topics included in the same sections and other sections that do not have an obvious link to photoperiodism (such as cotton evolution) which might be better described in a different manuscript.

Response: We thank this expert reviewer for the insightful analysis and constructive comments to improve the paper. Our responses to specific comments are as follows.

Some specific comments on the structure:

- Photoperiodism and circadian regulation affects many developmental processes, but it should be clear in the introduction, abstract and title that the focus of the review is on the vegetative to reproductive transition. The current title is very broad and the introduction section mentions different biological processes that are not relevant to the current study. A greater focus on the vegetative to reproductive transition would clarify for the reader what the focus of the paper is.

Response: As suggested, we have revised the title as well as the structure of the manuscript to provide a clearer narrative flow and better organization of topics related to photoperiodism and circadian responses. The new title is "Photoperiodism and Circadian Regulation from Vegetative to Reproductive Transition in Plants" to better highlight the focus on the transition from vegetative growth to reproduction.

In the revision, we added paragraphs in the Introduction to clarify basic concepts such as short-day and long-day requirements for flowering, as well as circadian gating, to provide a clear context for readers. Additionally, we have reorganized the relevant information to focus more closely on the genetic pathways underlying photoperiodism and circadian responses. Some unrelated topics and subjects including list of genes and their potential roles have been removed to enhance the clarity and coherence of the paper.

- Many concepts are not described adequately. It is not stated that Arabidopsis is a long-day plant and that these pathways have adapted to different photoperiods to promote flowering. The concept of circadian gating is not described, nor the temperature-mediated changes in the clock.

Response: We have revised the manuscript to address these issues. We have introduced the concepts of long-day, short-day, and day-neutral plants, clarifying that Arabidopsis is a facultative long-day plant with metabolic pathways adapted to different photoperiods for flowering promotion. Additionally, we have provided conceptual explanation of circadian gating and discussed temperature-mediated changes in the clock to ensure a more comprehensive coverage of the topic.

- There is a section combining epistasis and epigenetics, two different topics, that are covered very briefly with few examples and do not seem to be especially important for photoperiodism (These are facets across gene regulation). The inclusion of epistasis in this section is particularly unusual. The section then moves to description of polyploidization in cotton and a description of genes in this species. I understand this is an area of research expertise for the authors, but the topic here does not have special interest for photoperiodism and should probably be left out.

Response: We agree in principle with this assessment. Our intention was to make the readers aware of complicated gene and allelic interactions in complex crop genomes, in addition to transcriptional regulation by DNA methylation and chromatin modifications in response to seasonal changes and environmental cues. Compared to detailed studies in Arabidopsis, the underlying mechanisms and pathways are much less known in crop plants, many of which are polyploids or of polyploid origin. To facilitate a smooth transition, we added a paragraph to expand our understanding of epialleles such as *FLC* and *FWA*; the former is involved in response to cold winter via a vernalization process. Many other factors such as *CO* in cotton are epialleles, which result from intergenomic interactions in polyploid crops. The impact of epistasis is much less understood due to duplicate genes and networks in polyploid plants and crops. We hope the coverage of these topics will help introduce and develop translational research on crop plants. Although the findings are relatively limited compared to deep understanding in Arabidopsis, they should provide some guidelines for future studies on translating basic findings in Arabidopsis to crop improvement.

- The concluding remarks section mentions that an understanding of these pathways can lead to engineering yield improvement. It would be good to include some examples of crops for which this has been achieved to demonstrate the real-world outcomes of this research area.

Response: Thank you for the suggestion. “There are good examples of the genes in the circadian clock and photoperiodic flowering pathways that are associated with agronomic traits in crops. Rice *Hd1*¹, an ortholog of the Arabidopsis *CO*, controls expression of two florigen genes *Hd3a* and *RFT1*²⁻⁴ during the floral transition. *ZmCCT9* in maize is diurnally regulated and negatively regulates the expression of the florigen *ZCN8* through a CACTA-like TE insertion in *ZmCCT* promoter region⁵, resulting in reduction of photoperiod sensitivity, thus accelerating maize spread to long-day environments⁶. The circadian clock of cultivated tomatoes has slowed during domestication. *EMPFINDLICHER IM DUNKELROTEN LICHT1* (*EID1*) is an F-box protein that targets phytochrome A for degradation in Arabidopsis⁷. The *EID1* allele in cultivated tomatoes is under selection sweep and enhances plant performance under long day photoperiods presumably resulting from moving away from the equator⁸. Fruit yield in hybrid tomato is related to *SINGLE FLOWER TRUSS* (*SFT*)⁹, a homolog of *FT* in Arabidopsis. Moreover, epigenetic alteration of circadian clock genes including *CCA1* and *LHY* in plant hybrids are related to growth vigor in Arabidopsis¹⁰⁻¹² and maize^{13,14}. In cotton, *GhCOL2* is an epiallele, which may be selected during domestication to increase its expression and thereby reduce photoperiod sensitivity, helping spread worldwide cotton cultivation¹⁵. These examples will help us design strategies to optimize circadian input and output signals and improve flowering time and crop yield using genome- and gene-editing tools.” This is included in Conclusion.

Overall, I feel that the manuscript could be improved by clearly defining the topic and target audience, then rewriting to focus on these topics so they are most clearly understood, with a grounding in some of the underlying concepts. This will help readers to understand the relevance of each gene pathway and the broader context of how they fit in to plant biology.

Response: Thank you for your constructive comments. We have revised the manuscript to concentrate on the regulation of plant transitions from vegetative to reproductive stages, specifically highlighting photoperiodism, circadian rhythm, and epigenetic regulation. We have included additional context in the Introduction, Crop Flowering Time, and Conclusions to facilitate a clearer understanding and transition of these topics.

Minor comments

Line 41: Replace “Despite” with “although.

Response: The sentence along with other context was removed in the revised Introduction.

Line 42: “key regulators of floral transition are highly conserved”. This is poorly defined. I am familiar with work from wheat that shows the *CONSTANS* proteins, for example, have a less

important role in defining flowering time in this species. I believe this sentence should either be modified or removed.

Response: As suggested, the sentence was removed.

Line 84 – Sentence is incomplete, ends on “constitute”.

Response: We have completed the sentence as "...whose expression peaks in the evening; they constitute the central loop of the core oscillator."

Lines 113-116: These sentences seem out of place since they describe the conservation of regulators in different species. Was this the topic of the introduction?

Response: We moved these sentences to the top of the paragraph under the section of Photoperiodic Control of Flowering in Crop/Rice.

Line 203: “switch on flowering” could be defined in more technical terms.

Response: We replaced "switch on flowering" with "trigger flowering".

Lines 238 – 242: The genes mentioned here are from Soybean. Why is this information not grouped together in the section describing responses in soybean?

Response: Sorry for the confusion. We moved the first sentence to the section of soybean, while the sorghum PRR37 remains in the same place, followed by, “Similarly, clock components such as *OsPRR37/DTH7* and the EC components, also regulate rice flowering...”

Line 246: The evening complex in Arabidopsis is introduced mid-paragraph.

Response: We started the EC complex in a new paragraph.

Figure 2: What are the line graphs at the bottom of the figure?

Response: Sorry for the confusion. The line graphs at the bottom of the figure 2 indicate mRNA oscillation of *CO*, *CO/FKBPI2*, and *FT*, respectively. We clarified this in the legend and also in the text.

Reviewer #2 (Remarks to the Author):

This submission by Wang et al. provides a timely and comprehensive overview of the molecular mechanisms underlying photoperiodic regulation of flowering times in the model plant *Arabidopsis thaliana* and several economically-important crops. Overall the manuscript is well written. Below are suggestions for improvement.

Response: We appreciated this expert reviewer for the positive assessment of our work.

1. Title: it would be better to remove the word of crops because crops are plants.

Response: We have revised the title, “Photoperiodism and Circadian Regulation from Vegetative to Reproductive Transition in Plants.”

2. Lines 43-58: As these two paragraphs describe the mechanisms in *Arabidopsis thaliana*, not in other plants, please add the species name somewhere in the first paragraph.

Response: The section including Lines 43-58 was redundant in the later section and removed in the revision. The species name was spelled out in the first appearance.

3. Line 84:constitute?

Response: Sorry for the mistake. We have completed the sentence as "...whose expression peaks in the evening; they constitute the central loop of the core oscillator."

4. Lines 312-317: the function of *COOLAIR* in vernalization-mediated *FLC* repression has been questioned (e.g. DOI: 10.7554/eLife.84594 and other publications). Please rephrase this paragraph.

Response: As suggested, we have rephrased this paragraph. “One of the best studied epigenetic flowering events is known as vernalization, which refers to the induction of flowering after prolonged exposure to cold temperatures in winter, mediated by changes in the chromatin status of *FLC*¹⁶. *COLD INDUCED LONG ANTISENSE INTRAGENIC RNA (COOLAIR)*¹⁷ and *COLD ASSISTED INTRONIC NONCODING RNA (COLDAIR)*¹⁸ originated from the first intron and 3’ end of *FLC*, respectively. They have been proposed to facilitate *FLC* silencing by removing H3K36me3 from chromatin during vernalization, particularly under low-temperature conditions^{17,18}. However, recent research suggests that the *COOLAIR* is not required for vernalization. Support for this notion comes from the normal vernalization response observed in the CRT/DRE-binding factors (CBFs) *cbfs* mutants with reduced levels of *COOLAIR* induction; moreover, both *cbfs* and *FLC_{ΔCOOLAIR}* mutants show a normal vernalization response despite their inability to activate *COOLAIR* expression during cold¹⁹. This work also highlights the importance of CBFs in initiating *COOLAIR* expression during the early stage of vernalization. CBFs bind to CRT/DREs at the 3’-end of *FLC* and increase progressively during vernalization, promoting *COOLAIR* expression. As vernalization progresses, *FLC* chromatin shifts to an inactive state,

prompting CBF proteins to detach from the CRT/DREs in the *COOLAIR* promoter, thereby diminishing *COOLAIR* levels.”

5. A few typos. Line 91: actives to activates. Line 114: ...conserved IN duckweed... Line 135: diel to diurnal? Line 311: ...cold IN winter... Line 203: please italicize *OsMADS14* and *OsMADS15*.

Response: Thanks for careful proofreading. We have corrected these typos as suggested.

6. Lines 318-320: these two sentences are not consistent.

Response: We revised the sentence. "In *Arabidopsis*, the epiallele *FWA* encodes a flowering suppressor capable of specifically inhibiting the function of FT by directly binding to the FT protein."

References cited

1. Yano, M. et al. Hd1, a major photoperiod sensitivity quantitative trait locus in rice, is closely related to the *Arabidopsis* flowering time gene *CONSTANS*. *Plant Cell* **12**, 2473-2484 (2000).
2. Tsuji, H., Taoka, K.I. & Shimamoto, K. Regulation of flowering in rice: two florigen genes, a complex gene network, and natural variation. *Curr Opin Plant Biol* (2010).
3. Tamaki, S., Matsuo, S., Wong, H.L., Yokoi, S. & Shimamoto, K. Hd3a protein is a mobile flowering signal in rice. *Science* **316**, 1033-6 (2007).
4. Komiya, R., Ikegami, A., Tamaki, S., Yokoi, S. & Shimamoto, K. Hd3a and RFT1 are essential for flowering in rice. *Development* **135**, 767-74 (2008).
5. Yang, Q. et al. CACTA-like transposable element in *ZmCCT* attenuated photoperiod sensitivity and accelerated the postdomestication spread of maize. *Proc Natl Acad Sci U S A* **110**, 16969-74 (2013).
6. Huang, C. et al. *ZmCCT9* enhances maize adaptation to higher latitudes. *Proc Natl Acad Sci U S A* **115**, E334-E341 (2018).
7. Marrocco, K. et al. Functional analysis of *EID1*, an F-box protein involved in phytochrome A-dependent light signal transduction. *Plant J* **45**, 423-38 (2006).
8. Muller, N.A. et al. Domestication selected for deceleration of the circadian clock in cultivated tomato. *Nat Genet* **48**, 89-93 (2016).
9. Krieger, U., Lippman, Z.B. & Zamir, D. The flowering gene *SINGLE FLOWER TRUSS* drives heterosis for yield in tomato. *Nat Genet* **42**, 459-63 (2010).
10. Ni, Z. et al. Altered circadian rhythms regulate growth vigour in hybrids and allopolyploids. *Nature* **457**, 327-31 (2009).
11. Miller, M., Song, Q., Shi, X., Juenger, T.E. & Chen, Z.J. Natural variation in timing of stress-responsive gene expression predicts heterosis in intraspecific hybrids of *Arabidopsis*. *Nat Commun* **6**, 7453 (2015).
12. Ng, D.W. et al. A Role for CHH Methylation in the Parent-of-Origin Effect on Altered Circadian Rhythms and Biomass Heterosis in *Arabidopsis* Intraspecific Hybrids. *Plant Cell* **26**, 2430-2440 (2014).

13. Ko, D.K. et al. Temporal shift of circadian-mediated gene expression and carbon fixation contributes to biomass heterosis in maize hybrids. *PLoS Genet* **12**, e1006197 (2016).
14. Li, Z. et al. Temporal Regulation of the Metabolome and Proteome in Photosynthetic and Photorespiratory Pathways Contributes to Maize Heterosis. *Plant Cell* **32**, 3706-3722 (2020).
15. Song, Q., Zhang, T., Stelly, D.M. & Chen, Z.J. Epigenomic and functional analyses reveal roles of epialleles in the loss of photoperiod sensitivity during domestication of allotetraploid cottons. *Genome Biol* **18**, 99 (2017).
16. Whittaker, C. & Dean, C. The FLC Locus: A Platform for Discoveries in Epigenetics and Adaptation. *Annu Rev Cell Dev Biol* **33**, 555-575 (2017).
17. Swiezewski, S., Liu, F., Magusin, A. & Dean, C. Cold-induced silencing by long antisense transcripts of an Arabidopsis Polycomb target. *Nature* **462**, 799-802 (2009).
18. Heo, J.B. & Sung, S. Vernalization-mediated epigenetic silencing by a long intronic noncoding RNA. *Science* **331**, 76-9 (2011).
19. Jeon, M. et al. Vernalization-triggered expression of the antisense transcript COOLAIR is mediated by CBF genes. *Elife* **12**(2023).

REVIEWERS' COMMENTS:

Reviewer #1 (Remarks to the Author):

Thank you for carefully addressing my suggestions on content of this review article.